# Screening of Lactic Acid Bacteria with Inhibitory Activity against ETEC K88 as Feed Additive and the Effects on Sows and Piglets

**DOI:** 10.3390/ani11061719

**Published:** 2021-06-09

**Authors:** Weiwei Wang, Hao Ma, Yajie Zhu, Kuikui Ni, Guangyong Qin, Zhongfang Tan, Yanping Wang, Lei Wang, Huili Pang

**Affiliations:** 1Henan Key Lab Ion Beam Bioengineering, School of Agricultural Sciences, Zhengzhou University, Zhengzhou 450052, China; wangwei508@foxmail.com (W.W.); mahaoworks@foxmail.com (H.M.); swwlwww@126.com (Y.Z.); qinguangyong@zzu.edu.cn (G.Q.); tzhongfang@zzu.edu.cn (Z.T.); wyp@zzu.edu.cn (Y.W.); 2School of Physics and Microelectronics, Zhengzhou University, Zhengzhou 450052, China; 3College of Grassland Science and Technology, China Agricultural University, Beijing 100193, China; nikk@cau.edu.cn; 4Academy of Animal Science and Veterinary Medicine, Qinghai University, Xining 810016, China; bingzhi213608@163.com

**Keywords:** lactic acid bacteria, ETEC K88, sow, piglet, antioxidant capacity, immune indexes

## Abstract

**Simple Summary:**

Numerous reports have suggested that lactic acid bacteria (LAB), which are important probiotics, can protect animals against pathogen-induced injury and inflammation, regulate gut microflora, enhance digestive tract function, improve animal growth performance, and decrease the incidence of diarrhea caused by enterotoxigenic (ETEC) that expresses K88. This research selected *Lactobacillus* (*L*.) *reuteri* P7, *L. amylovorus* P8, and *L. johnsonii* P15 with good inhibition against ETEC K88 and excellent probiotic properties screened from 295 LAB strains isolated from fecal samples from 55 healthy weaned piglets for a study on feeding of sows in late pregnancy and weaned piglets. Feed supplementation with these three strains improved reproductive performance of sows and growth performance of piglets, decreased the incidence of diarrhea in piglets, and increased the antioxidant capacity of serum in both sows and piglets. Therefore, *L. reuteri* P7, *L. amylovorus* P8, and *L. johnsonii* P15 might be considered as potential antibiotic alternatives for further study.

**Abstract:**

Enterotoxigenic *Escherichia coli* (ETEC), which expresses K88 is the principal microorganism responsible for bacterial diarrhea in pig husbandry, and the indiscriminate use of antibiotics has caused many problems; therefore, antibiotics need to be replaced in order to prevent diarrhea caused by ETEC K88. The objective of this study was to screen excellent lactic acid bacteria (LAB) strains that inhibit ETEC K88 and explore their effects as probiotic supplementation on reproduction, growth performance, diarrheal incidence, and antioxidant capacity of serum in sows and weaned piglets. Three LAB strains, P7, P8, and P15, screened from 295 LAB strains and assigned to *Lactobacillus* (*L.*) *reuteri*, *L. amylovorus*, and *L. johnsonii* with high inhibitory activity against ETEC K88 were selected for a study on feeding of sows and weaned piglets. These strains were chosen for their good physiological and biochemical characteristics, excellent exopolysaccharide (EPS) production capacity, hydrophobicity, auto-aggregation ability, survival in gastrointestinal (GI) fluids, lack of hemolytic activity, and broad-spectrum activity against a wide range of microorganisms. The results indicate that LAB strains P7, P8, and P15 had significant effects on improving the reproductive performance of sows and the growth performance of weaned piglets, increasing the activity of antioxidant enzymes and immune indexes in both.

## 1. Introduction

Aside from environmental and disease factors, the slow growth of fattening pigs has a lot to do with sow feeding during gestation and the health of weaned piglets. When the foundation of health is strong during gestation and the newborn piglets are big after birth, the piglets will win at the starting line, as the saying goes, if a newborn weighs 50 or 100 g, and at weaning weighs 500 g, ultimately it will weigh 5000 g [1]. Enterotoxigenic *Escherichia coli* (ETEC) that express K88 fimbriae are the principal microorganisms responsible for bacterial diarrhea in weaned piglets, which results in an economic loss of 26% [2,3]. Antibiotics are used to treat a variety of bacterial infections or suppress infections by disease-causing microorganisms, including various types of diarrhea, but unfortunately, long-term and heavy use of antibiotics has caused many problems, such as dysbiosis of animal intestinal microbiota, drug residues, and environmental pollution [4]. The use of antibiotics as growth promoters in diets for weaned pigs in the European Union has been banned since 1 January 2006 (European Community Regulation 1831/2003) [5]. With announcement No. 194 of the Ministry of Agriculture and Rural Affairs, China has totally banned the use of antibiotics in feed since 1 January 2020 because of the emergence of antimicrobial resistance in animals and humans. Some studies have shown that zinc oxide (ZnO), organic acids, feed enzymes, prebiotic oligosaccharides, and clay minerals can improve post-weaning diarrhea associated with ETEC, but the impact on the environment, risks associated with microbial resistance, and difficulty of material preparation must be considered [6,7,8]. As a consequence, antibiotic growth promoters must be replaced in order to enhance growth performance and prevent diarrhea caused by ETEC K88.

Lactic acid bacteria (LAB), which are among the most common and typical probiotics, exist widely in nature and are isolated mainly from fermented foods, soil, and animal gut and fecal matter [9,10]. Several studies have confirmed that LAB can enhance intestinal barrier function of weaned piglets. *Lactobacillus* (*L.*) *plantarum* protected the intestinal barrier function of weaned piglets against ETEC challenge [11]; *L. reuteri* reduced ETEC K88 in the intestines of weaned piglets [12], affected the abundance of pathogenic bacteria, and particularly reduced the abundance of ETEC K88 [13]; *L. salivarius* enhanced growth performance and decreased the incidence of diarrhea caused by ETEC K88 [14]; and *L. casei* OLL2768 was a good candidate against intestinal inflammatory damage induced by ETEC [15]. In summary, above studies show that LAB can inhibit pathogenic bacteria, including intestinal imbalance caused by ETEC K88. Additionally, by immunizing gestating sows, newborn piglets receive maternal antibodies against ETEC from the colostrum, providing early and effective immune protection against diarrhea caused by ETEC infection.

As the livestock industry is important worldwide, the demand for LAB in the farming industry is large, while the market for feeding LAB is still in the primary stage and the number of commercialized LAB products is relatively small at present. Therefore, this study aimed to isolate pig-derived LAB strains and screen out safe and effective LAB against ETEC K88 by comprehensively evaluating their in-vitro and in-vivo beneficial properties for both sows and piglets, and provide LAB resources for the development of complex microecological preparations for pigs.

## 2. Materials and Methods

### 2.1. Sample Collection and LAB Isolation

Fifty-five fresh fecal samples from healthy weaned piglets were collected randomly at a pig farm in Xinxiang, Henan, China, in September 2016. Uncontaminated fresh fecal from the upper part of the fecal pellet naturally expelled by each healthy piglet were collected with sampling spoons and samples were placed in sterile EP tubes and marked, and tubes were immediately transported to the laboratory for analysis by cold chain in a sample box with enough dry ice. It took no more than 2 h from sampling to analysis. Fecal samples were diluted in a gradient with distilled water, and then coated on de Man, Rogosa, and Sharpe (MRS) agar (Merck, Darmstadt, Germany) before being incubated at 37 °C for 48 h under anaerobic conditions to culture single LAB colonies in fresh fecal samples and isolate them. Strains with colonies round, raised or flat, creamy white or slightly yellow, moist, medium size, neatly edged, and Gram stain positive and catalase reaction negative, were tentatively identified as LAB and kept at −80 °C for further tests.

### 2.2. Screening of LAB with Inhibitory Activity against ETEC K88

The well diffusion technique was performed with reference to the technique of Sirichokchatchawan et al. [16]. Colony of the target bacteria ETEC K88 was put in nutrient liquid medium and incubated at 37 °C with 180 rpm for 12 h, after the concentration reached 1 × 10^8^ CFU (Colony-Forming Units)/mL, 200 µL bacterial solution was taken and coated on NA (Nutrient Agar) plates, and 200 µL 16 h cultures of different LAB strains (1 × 10^8^ CFU/mL) were placed separately in the hole on NA plates punched by hole puncher (diameter 10.00 mm), and the hole with uninoculated MRS broth and penicillin (pc) were used as negative and positive control, respectively. The target bacteria ETEC K88 was purchased from the China Veterinary Culture Collection Center (CVCC). LAB strains with relatively larger inhibition zone diameters were selected for determination of physiological and biochemical characteristics.

### 2.3. Physiological and Biochemical Characteristics of Selected LAB Isolates

Physiological and biochemical characteristics such as pH, salt and temperature tolerance of selected LAB strains were all determined in MRS broth. After each single LAB colony was picked and added into 20 mL sterile MRS liquid medium, it was grown at 37 °C for 16 h and adjusted to an optical density (OD) of 0.8 at 600 nm with sterile water, then 100 µL different LAB bacteria liquid was separately mixed with 9.9 mL MRS broth. The acid and alkali resistance of the LAB strains were evaluated in MRS broth with different pH values (3.0, 3.5, 4.0, 4.5, 8.0, 9.0 and 10.0) and incubated for 7 days at 37 °C. Salt tolerance was determined in MRS containing 3.0 and 6.5% NaCl, incubated for 48 h at 37 °C. Temperature tolerance was measured in MRS broth and incubated under temperatures of 4, 10, 40, 45, and 50 °C for 7 days, respectively. After incubation, the growth rates of LABs were assayed using the turbidimetry method by measuring absorbance values at OD600 nm combined with visual turbidity, with a sterile MRS medium without inoculating as controls. Among them, OD600 value of control would be recorded as 0, compared to the control, 0 ≤ OD600 ≤ 0.2 would be considered to not growth and recorded as “-”, 0.2 < OD600 ≤ 0.6 as weakly growth and “w”, and OD600 > 0.6 growth and “+” [17], respectively.

### 2.4. Exopolysaccharide (EPS) Produced Capacity of Selected LAB Isolates

The capacity to produce EPS of selected LAB strains was evaluated using the methodologies described by Bachtarzi et al. [18] and Pacularu-Burada et al. [19] with some modifications. Briefly, the biomass of a single colony of LAB grown on MRS agar (MRS supplemented with 50 g/L sucrose and 5 g/L glucose), after incubation at 37 °C for 72 h, was visually inspected to select slimy and mucoid LAB colonies that formed wires to the touch with a sterile loop for further tests. A volume of 180 µL of each selected LAB strain previously reactivated in MRS broth was then aseptically inoculated in a sterile glass tube containing 9 mL of modified MRS broth (2% LAB) and incubated at 37 °C for 72 h under aerobic conditions; afterwards, the supernatant was collected by centrifugation and its absorbance was measured at 490 nm; results were expressed as gram of total sugars per liter of medium.

### 2.5. Cell Surface Hydrophobicity and Auto-Aggregation of Selected Strains

The adhesion ability of selected LAB strains is reflected by cell surface hydrophobicity and auto-aggregation, and these two characteristics were tested using methods described by Wang et al. [20]. Strains with high cell surface hydrophobicity and auto-aggregation were selected for 16S rRNA gene analysis.

### 2.6. 16S rRNA Gene Analysis of Selected Strains

Selected LAB strains were identified by genetic analysis using PCR and 16S rRNA gene sequencing. The universal primers 27 F (5′-AGAGTTTGATCCTGGCTCAG-3′) and 1492 R (5′-GGTTACCTTGTTACGACTT-3′) were used for PCR amplification of the 16S rRNA gene. The successful amplification was analyzed by a sequencing service (MGI Tech Co., Ltd., Beijing, China), sequence similarities of each contig were examined by searching their homologies in the GenBank database using BLAST (http://www.ncbi.nlm.nih.gov, in 4 September 2020), and the phylogenic tree was constructed by the neighbor-joining method with MEGA-X software.

### 2.7. Antimicrobial Activity of Selected LAB Isolates

*Bacillus subtilis* ATCC 19217^T^, *Pseudomonas aeruginosa* ATCC 15692^T^, *Salmonella enterica* ATCC 43971^T^, *Escherichia coli* ATCC 11775^T^, *Listeria monocytogenes* ATCC 51719^T^, *Staphylococcus aureus* ATCC 6538^T^, *Micrococcus luteus* ATCC 4698^T^, and *Escherichia coli* ATCC 11775^T^ were used as indicator bacteria to assess the antimicrobial activity of selected LAB isolates by agar well diffusion technique, and all indicator bacteria were cultured on nutrient liquid medium at 37 °C with 180 rpm for 12 h until the concentration reached to 1 × 10^8^ CFU/mL, then took 200 µL bacterial solution to coat on each NA plates. LAB strains were cultured on MRS liquid medium under 37 °C for 16 h to make the concentration to 1 × 10^8^ CFU/mL. Susceptible test was determined by the inhibition zone method, and the specific operation was the same as described in Section 2.2, sensitivity was positively correlated with the diameter of the zone of inhibition, while uninoculated MRS broth was used as control [21]. The inhibition spectrum and inhibition ability of each LAB strain were determined according to the inhibition zone diameter, and strains with relatively larger inhibition zone diameters were selected for further studies.

### 2.8. Survival of Selected LAB Strains in Bile Salt and Simulated Gastrointestinal Fluids

MRS solution containing 0.5% (*w*/*v*) bile salt was used to assess bile salt tolerance. Each LAB isolate was incubated in bile salt environments for 2, 4 and 6 h, respectively, and then incubated at 37 °C for 20 h, with the biomass being measured by OD 600 nm. The viability of selected strains in simulated gastric fluid (SGF) and simulated intestinal fluid (SIF) was determined according to a method described by Massounga Bora et al. [22] with modifications. Briefly, for SGF, 3.5 g/L pepsin was suspended in 0.2% *w*/*v* sterile NaCl solution and pH was adjusted to 2.0, making the total volume of the solution up to 100 mL with distilled water; for SIF, 1 g/L trypsin, suspended 18 g/L bile salt from ox and 11 g/L NaHCO_3_ into 0.2% *w*/*v* sterile NaCl solution, adjusted the pH of the solution to 6.8 and brought the total solution volume to 100 mL with distilled water, filtered through a 0.22 µm filter membrane, and the criterion for strain selection was more colonies on the plate after incubation.

### 2.9. Assessment of Antibiotic Susceptibility of Selected LAB Isolates

Antibiotic susceptibility tests of the isolates against 10 antibiotics carbenicillin, cefamezin, ampicillin, gentamicin, norfloxacin, clindamycin, penicillin, erythromycin, chloramphenicol, and amikacin were conducted by the disc diffusion method in a nutrient broth medium according to the guidelines of the Clinical and Laboratory Standards Institute [23]. Ten antibiotics were added to MRS agar medium separately and 4 LAB isolates to be test were coated on each medium, and the growth of LABs on the medium was observed after 24 h incubation under 37 °C, respectively. Disc agar diffusion method was used to test antibiotics susceptibility, and susceptibility to antibiotics was expressed as the diameter of the inhibition zone in millimeters, determined the drug susceptible (S), moderate susceptible (I) and drug resistance (R) according to the instructions.

### 2.10. Hemolytic Activity of Selected LAB Isolates

Hemolytic activity of LAB isolates assay was determined according to the description in Wang et al. [20], and *Staphylococcus aureus* ATCC 6538^T^ with hemolytic properties was used as positive control. Strains that tested free of hemolysis would be used for animal testing.

### 2.11. Animals, Diets, and Experimental Design

The experiment was approved by the Institutional Animal Care and Use Committee at Zhengzhou University, Zhengzhou, Henan, China (15 July 2016). A total of 36 sows all from the same breed Duroc × Yorkshire × Landrace of similar weight and same litter size at 81–85 days pregnant were selected from a pig farm (Xinxiang, Henan, China) and randomly divided into 2 groups: the control group with basal diet, with separate formulas during pregnancy and lactation (Table 1), and the test group with 6% mixed selected LAB strains in equal amounts, at 2 × 10^8^ CFU per gram (CFU/g), as supplementation to the basal diet. Each treatment had 3 replicates and 6 sows per pen, the trial lasted 55 days until the end of lactation, and the sows were also fed with LAB after production until weaning. The temperature and relative humidity of the farrowing room were strictly controlled at 19 ± 1 °C and 60%. For piglet trials, 60 weaned 24-day-old piglets (Duroc × Yorkshire × Landrace, 12.0 ± 0.12 kg body weight) from the control sow group, with a similar health status, were randomly assigned to control and antibiotic groups balanced for sex and weight, and 30 weaned piglets from the LAB sow group were assigned to the LAB group using the same method. Piglets were fed 3 diets as follows: (1) control group: basal diet (Table 1); (2) LAB group: basal diet + 6% LAB (same as sow experiment); and (3) antibiotic group: basal diet + 150 mg/kg of aureomycin. Each treatment had 3 replicates and 10 piglets per pen, who were kept in a temperature-controlled nursery room (25 ± 1 °C). Piglets fed LAB after weaning and the trial lasted for 4 weeks. The basal diet was formulated according to the nutrient requirements recommended by the National Research Council (2012). The regular vaccination program at the farm for sows and piglets were as follow: for sows, classical swine fever (live, tissue culture origin), swine parvovirus (inactivated, strain WH-1), and porcine reproductive and respiratory syndrome (live, strain CH-1R), SlV inactivated and swine foot and mouth disease (Type O) vaccine (inactivated (II)) were vaccination 30, 20 and 10 days before mating; swine atrophic rhinitis (inactivated), pseudorabies (live, strain Bartha-K61), porcine reproductive and respiratory syndrome vaccine (live, strain CH-1R) on 40–42 and 25–30 days before production; swine parvovirus (inactivated, strain WH-1), classical swine fever vaccine (live, tissue culture origin) on 25 and 28–40 days after production, respectively. For piglets, pseudorabies (live, strain Bartha-K61) and swine atrophic rhinitis (inactivated), porcine reproductive and respiratory syndrome vaccine (live, strain CH-1R) on 15–18 days and 20–28 days, respectively. The farm environment was cleaned and disinfected twice a month, and mice, mosquitoes and flies were killed at the same time, while the pigpen were cleaned and disinfected twice a week. All sows and piglets had free access to feed and water throughout the experimental period.

### 2.12. Data and Sample Collection of Sows and Weaned Piglets

For each sow, reproductive performance, including litter size at birth (total born, born alive, weak piglets, and stillborn piglets), birth weight (per litter and individual), live litter rate, weak litter rate, weaning survival rate, estrus rate of 1–7 and 8–14 days, and conception rate during estrus were recorded. Three sows from each replicate were randomly selected to determine serum antioxidant capacity and immune indexes at the end of the feeding trial; serum was obtained by centrifugation at 6000× *g* for 15 min at 4 °C, and stored at −80 °C until analysis. Serum antioxidant parameters including total superoxide dismutase (T-SOD), total antioxidant capacity (T-AOC) and malondialdehyde activity (MDA) were measured using methods of Hydroxylamine, ABTS and TBA, serum immune indexes as tumor necrosis factor-α (TNF-α) and interferon-γ (IFN-γ) were both analyzed by ELISA method, and immunoglobulins (IgG, IgM and IgA) were determined by Turbidimetric method, respectively. All assays were performed by kits provided by Nanjing Jiancheng Bioengineering Institute (Nanjing, Jiangsu, China).

During the entire experimental period, daily feed offered and remaining were weighed on an electronic scale (HY-602; Jinxuan, Zhejiang, China) and the amounts were recorded to monitor average daily feed intake (ADFI). All piglets were individually weighed per week and their growth performance was assessed in terms of average daily gain (ADG) and feed: gain ratio (F:G), calculated by dividing ADFI by ADG. Diarrheal incidence in all piglets was checked at least once a day.

Diarrhea rate (%) = (total number of piglets with diarrhea in each pen)/(number of piglets in each pen × trial days) × 100. The method of determining piglet serum antioxidant capacity was the same as that for sows, also 3 piglets from each replicate were randomly selected for index determination.

### 2.13. Statistical Analysis

Data were obtained from at least 3 replicate experiments. Microsoft Excel 2019 software was used to process the data, and values from all experiments were expressed as mean ± standard deviation (SD). Statistical analysis was performed using SPSS statistical package 17.0 (SPSS Inc., Chicago, IL, USA) by one-way analysis of variance (ANOVA) to compare the effects of different LAB on EPS production capacity, cell surface hydrophobicity, auto-aggregation ability, survival after simulated GIT exposure, reproductive performance of sows, and immune indexes of serum in sows and weaned piglets, and Tukey’s test was used for comparisons at the 5% significance level. In addition, paired-sample *t*-tests were employed to compare the growth performance of weaned piglets, as well as the antioxidant capacity of serum in sows and weaned piglets; 0.01 < *p* < 0.05 indicated significant difference and *p* < 0.01 extremely significant difference, marked with * and **, respectively.

## 3. Results

### 3.1. Isolated LAB Strains from Piglet Fecal Samples

A total of 295 strains preliminarily identified as LAB with positive Gram stain and negative catalase reaction were isolated from fecal samples from 55 healthy weaned piglets by selective medium using culture method, of which 279 strains were homo-fermentative (data not shown).

### 3.2. Inhibitory Activity of LAB Strains against ETEC K88 Isolated from Piglet Fecal Samples

The inhibitory activity of isolated LAB strains against ETEC K88 is shown in Table 2. As can be seen, 20 strains (P1 to P20; strains were uniformly labeled as P series for better documentation and preservation) had an inhibition zone of at least 14.00 mm (including that of the hole, puncher 10.00 mm), and among these strains, P1, P3, P4, P5, P7, P8, P10, P11, P14, P15, P16, P17, P18, and P20, with an inhibition zone of more than 18.00 mm, were further evaluated for their physiological and biochemical characteristics. The remaining strains were not listed and eliminated from the primary screen because of their poor inhibition.

### 3.3. Physiological and Biochemical Characteristics of Selected LAB Isolates

Physiological and biochemical characteristics of selected LAB isolates are shown in Table 3. All representative strains were able to grow in 3.0 and 6.5 (*w*/*v*, %) NaCl, at pH 4.0, 4.5, 8.0, 9.0, and 10.0, at 10, 40, and 45 °C, of which 9 strains (P1, P3, P5, P7, P8, P11, P15, P16, and P20) weakly grew at 4 °C, and 3 strains (P8, P14, and P16) had a good growth profile at 50 °C. Moreover, only P15 and P16 grew at pH 3.0, and other than P10, which was hetero-fermentative, all strains were homo-fermentative. Therefore, 9 strains (P1, P3, P5, P7, P8, P11, P15, P16, and P20) with relatively good temperature, salt, acid, and alkali tolerance were screened for subsequent experiments.

### 3.4. EPS Production Capacity of Selected LAB Isolates

EPS production capacity of selected LAB isolates is shown in Figure 1. It can be seen that after 72 h incubation at 37 °C, significant differences in EPS produced by these 9 isolates were observed, and the EPS quantity ranged between 12.07 and 16.86 g/L. Isolate P8 produced the highest content of EPS at 16.86 g/L, while P20 produced the lowest at 12.07 g/L. Consequently, strains P3, P5, P7, P8, P11, P15, and P16 with the relatively strongest EPS-producing ability were selected for further research.

### 3.5. Hydrophobicity and Auto-Aggregation Ability of Selected LAB Isolates

The adhesion ability of 7 selected LAB isolates is shown in Figure 2. Among these strains, P7, P8, P15, and P16 had a hydrophobic rate of more than 45.40%, of which P15 had the highest at 69.07%, and P11 had the lowest at only 7.17% (Figure 2a). The auto-aggregation ability of isolates is shown in Figure 2b. After 8 h treatment, the auto-aggregation rate for P7, P8, P15, and P16 was higher than 59.90%, and for P11 was only 15.13%; P15 also showed the highest ability at 78.00%. Based on the results presented in Figure 2, isolates P7, P8, P15, and P16 were chosen for further study.

### 3.6. 16S DNA Sequence Analysis

Figure 3 shows phylogenetic trees constructed using P7, P8, P15, and P16 strains based on evolutionary distances determined by the neighbor-joining method, and these 4 strains were set in the *Lactobacillus* cluster, while strains P7 and P16 were identified as *L. reuteri* with 100% bootstrap values, and P15 and P8 were assigned to *L. johnsonii* and *L. amylovorus* species, respectively, both supported by 100% sequence similarities and bootstrap values.

### 3.7. Antimicrobial Spectrum Test of Selected LAB Isolates

In this research, P7, P8, P15, and P16 were selected to assess antimicrobial potential (Table 4), and all 4 strains showed extremely strong inhibition against *Escherichia coli* ATCC 11775^T^, with an inhibition zone more than 18.00 mm in diameter and moderate ability against *Staphylococcus aureus* ATCC 6538^T^, *Bacillus subtilis* ATCC 19217^T^, *Micrococcus luteus* ATCC 4698^T^, and *Salmonella enterica* ATCC 43971^T^ with an inhibition zone diameter of 14.00 to 22.00 mm. Among these, P16 had no inhibitory effect against *Pseudomonas aeruginosa* ATCC 15692^T^ and an inhibition zone of only 10.00 to 14.00 mm in diameter with *Listeria monocytogenes* ATCC 51719^T^, which was the same for P8 with *Pseudomonas aeruginosa* ATCC 15692^T^. Moreover, the inhibition zone diameter of P15 against all pathogenic strains was at least 18.00 mm.

### 3.8. Survival of Selected LAB Isolates after Simulated Gastrointestinal Tract (GIT) Exposure

Figure 4a shows the 0.5% bile salt resistance of 4 selected LAB isolates cultivated at 37 °C. The OD values of all 4 test LAB isolates were significantly different after being treated with 0.5% bile salt for 2, 4, and 6 h. All isolates had certain vigor after 2 h of incubation at 37 °C, of which the highest was P15 with 1.381. After 4 h incubation, P8 and P15 still had certain vigor, and P15 had an OD value of 1.141. After 6 h of treatment, the OD values of P8 and P15 were 0.815 and 0.756, respectively, while other strains showed no activity.

Figure 4b illustrates the viable count (log CFU/mL) of 4 LAB strains during GIT exposure. After 3 h incubation in SGF, cell numbers of P8 and P16 were significantly decreased to 6.30 and 4.70 log CFU/mL from a starting point of 8.00 log CFU/mL; P7 decreased to 7.28 and P15 to 7.82 log CFU/mL. After 4 h incubation in SIF, the populations of P8 and P16 were 5.30 and 4.16 log CFU/mL, and the survival ratio was 66.2 and 52.3% in GIT, respectively, and after 4 h incubation in SIF, the populations of P7 and P15 were 5.67 and 7.23 log CFU/mL and the survival ratio were 71.1 and 89.8% in GIT, respectively.

For comprehensive in-vivo investigation of resistance to simulated GIT and safety evaluation, strains P7, P8, and P15 were selected.

### 3.9. Assessment of Antibiotic Susceptibility

Table 5 shows antibiotic susceptibility of 4 selected LAB isolates. As can be seen, all LAB strains displayed susceptibility to carbenicillin, cefamezin, clindamycin, and chloramphenicol and resistance to amikacin and norfloxacin. P7 and P15 showed intermediate resistance to gentamicin and penicillin, whereas P8 and P16 were quite the opposite.

### 3.10. Hemolytic Activity of Selected LAB Isolates

There was no transparent hemolysis circle around the 4 selected LAB strains compared with positive control *Staphylococcus aureus* ATCC 6538^T^, which means that the LAB strains had no hemolytic activity (data not shown).

### 3.11. Reproductive Performance of Sows, Growth Performance of Piglets, and Incidence of Diarrhea

Effects of LAB on the reproductive performance of sows are presented in Table 6. Sow feeding experiments showed that supplementation of feed with LAB resulted in increases in the following parameters: number of piglets at birth and birth weight per litter (both *p* < 0.05), conception rate during estrus (especially significant at *p* < 0.01), and lower numbers of weak piglets (*p* < 0.05) compared with control.

The effects of 3 selected LAB strains and antibiotics on growth performance and diarrhea rate of weaned piglets are shown in Figure 5. During the entire 4-week experimental period, LAB increased final body weight (*p* < 0.05) of weaned piglets compared to control and antibiotic groups (Figure 5a). Figure 5b shows that the incidence of diarrhea was lower in LAB (*p* < 0.05) and antibiotic (*p* = 0.071) groups than that in the control group. Compared with the control group, ADG, ADFI, and the efficiency of feed utilization of piglets in the LAB treatment group were significantly increased (*p* < 0.05) at 400.9 g, 664.3 g, and 1.66, respectively, and in the antibiotic group, these 3 indicators were greatly increased (*p* < 0.05), to 387.0 g, 661.0 g, and 1.71; there was no significant difference between the LAB and antibiotic treatment groups (Figure 5c–e).

### 3.12. Antioxidant Capacity and Immune Indexes of Serum in Sows and Weaned Piglets

Figure 6 shows the antioxidant capacity of serum in sows and weaned piglets. T-SOD (*p* = 0.248) and T-AOC (*p* = 0.399) concentrations of sows showed no significant difference between LAB and control groups (Figure 6a,b), while MDA concentration in the LAB group was lower compared with the control (*p* < 0.05) (Figure 6c). As for piglets, T-SOD concentrations were significantly higher (*p* < 0.05) and the MDA concentration was lower in both LAB and antibiotic treatment groups compared to control (*p* < 0.05) (Figure 6d,f). There was no significant difference between the LAB and antibiotic treatment groups, and for T-AOC concentration, there was no difference among three groups (*p* > 0.05) (Figure 6e).

The immune indexes of serum in sows and weaned piglets are listed in Table 7. In the LAB group of sows, TNF-α and IgA were higher compared with the control (*p* < 0.05), while IFN-γ, IgG and IgM concentrations showed no significant differences between LAB and control groups. For piglets, TNF-α and IgA concentrations were significantly higher (*p* < 0.05) compared to the control and antibiotic groups, while IgG was higher in both LAB and antibiotic treatment groups compared to the control (*p* < 0.05), but there were no significant differences between the LAB and antibiotic treatment groups. For IFN-γ and IgM concentrations, there were no differences among the 3 groups (*p* > 0.05).

## 4. Discussion

Diarrhea in piglets is a clinical sign of early infection by ETEC and other enteropathogens, including some bacteria as *Clostridium perfringens*, and *Salmonella* sp., viruses and parasites, and *Escherichia coli* is a common bacterium inside and outside the farm environment; after the lactation period, the piglets’ immune system had developed to a certain degree, and they basically adapted to it and rarely got sick. The diarrhea problem in piglets is closely related to their intestinal bacterial colonization and immune system development during the gestation and lactation periods [24], as well as changes occurring in diet and environment, morphological of intestine, and immunological and enzymatic during the weaning period [25]. Intestine of piglet is sterile at birth and, through nursing and contact with the environment, such as the farrowing bed, foreign microbiota gradually enter the intestine and they establish their own intestinal microbiota, maintaining the micro-ecological balance of the intestine [26]. Thus, as one of the most important factors in relation to piglet health, good inherited immunity from sows has a positive effect on the growth performance of piglets [27]. Colostrum provides energy and warmth, thereby increasing body temperature and viability [28]; moreover, colostrum rich in maternal antibodies and other immune factors, such as IgG, IgA, IgM, leukocytes and selenium, is also able to offer favorable immune protection to piglets [29]. In addition, immunity from milk, which contains a great deal of sIgA and other immune factors during lactation, can protects piglets until weaning [30]. Currently, antibiotics and alternatives to antibiotics such as ZnO, organic acids, probiotics, symbiotics, antimicrobial peptides, and spray-dried plasma (SDP) are used to relieve diarrhea in piglets, and a hygienic breeding environment and good feeding management are also important factors [31]. In addition to the importance of vaccines, LAB play an important role in livestock and poultry farming. Homologous LAB that come from the animals themselves are more adaptable to the environment and can colonize and work quickly, whereas many strains of commercially available microbial agents are harmful to the environment, which reduces the likelihood that they will colonize and function in vivo [32].

In the present study, 295 LAB strains were isolated from fecal samples from healthy weaned piglets, and 20 strains showed antagonistic activity against ETEC K88 by agar well diffusion technique. Among these strains, 14 that had an inhibition zone more than 18.00 mm in diameter were selected for further studies. In a study by Sirichokchatchawan et al. [16], the strain *L. plantarum* showed antagonistic activity against *Escherichia coli*; an inhibition zone diameter of 6.00–9.00 mm was considered to indicate weak inhibition (+), 10.00–13.00 mm was considered intermediate inhibition (++), 14.00–16.00 mm was considered strong inhibition (+++), and 17.00 mm or more was considered very strong inhibition (++++) by the agar well diffusion method (including the hole puncher, 6.00 mm). Pazhoohan et al. [33] reported that *Lactobacillus* showed antagonistic activity against ETEC with an inhibition zone diameter of 27.00 mm by the same method. Wang et al. [20] screened the strain *L. plantarum* subsp. *plantarum* ZA3, which showed antagonistic activity against ETEC K88 with an inhibition zone 23.00 mm in diameter by this method.

The normal pH value of the gastric juice is about 2.0, and after eating the pH rises to 3.5 because the gastric juice is diluted, thus only LAB with good tolerance to the gastric acid environment can survive in the stomach and be transported to the intestine to play a beneficial role. The typical bile salt concentration in the body is 0.3–0.5%. When LAB from food enter the small intestine through gastric digestion, they are exposed to a high-bile-salt living environment. Therefore, good bile salt tolerance is also one of the criteria for screening whether probiotic strains can perform beneficial functions in the gastrointestinal tract [9]. All representative strains in this study were able to grow in 3.0 and 6.5 (*w*/*v*, %) NaCl at pH 4.0, 4.5, 8.0, 9.0, and 10.0 and at 10, 40 and 45 °C. Other than P10, all strains were homo-fermentative; better still, P15 and P16 could grow at pH 3.0. All of these characteristics indicate that the selected strains can meet the requirements for growth in relatively extreme environments and have great potential for practical application.

EPS is a saccharide compound secreted by microorganisms during growth and metabolism that can protect the microorganisms [34]. Bachtarzi et al. [18] screened *L. plantarum* LBIO28 with high EPS production used in the production of dairy products, and Pacularu-Burada et al. [19] reported EPS quantity in the range of 11.307 to 23.193 mg/mL through phenol-sulfuric acid assay. In the present study, EPS quantity ranged from 12.07 to 16.86 g/L for these strains using the same test method; it is speculated that selected strains had exclusively good EPS production ability, and significant differences among isolates may depend on the strain type and the growth environment.

Cell surface properties including hydrophobicity and auto-aggregation are important for LAB [35]. P7, P8, P15, and P16 in this research had a hydrophobic rate of more than 45.40%, the highest of which was 69.07% for P15, and an auto-aggregation rate higher than 59.90% after 8 h treatment, and P15 showed the highest ability at 78.00%. These results are in agreement with previous studies showing a positive correlation between the surface hydrophobicity of the isolate and its self-aggregation ability [36], and the same findings also appeared in the research of Somashekaraiah et al. [37], who reported that *L. brevis* MYSN 98 showed the highest hydrophobic activity (77.82%) and auto-aggregation ability (78.95%).

Identifying screened strains at the species level is necessary to understand the habits, metabolic patterns, and pathogenicity of the bacteria [38]. Selected strains P7 and P16 were identified as *L. reuteri*, and P15 and P8 as *L. johnsonii* and *L. amylovorus*, respectively. *L. reuteri* is a naturally occurring microbiota in the intestinal tract of animals; it can produce bacteriocins to inhibit the growth of intestinal pathogenic bacteria, and at the same time has the ability of adhesion, colonizing in the intestinal tract and competing with pathogenic bacteria for adhesion sites of intestinal epithelial cells, thus reducing the risk of invasion by pathogenic bacteria [39]. *L. johnsonii* has good probiotic effects, which can improve the growth performance of dogs, and reduce the colonization of pathogenic bacteria [40]. *L. amylovorus* can inhibit the growth of pathogenic bacteria by producing large amounts of hydrogen peroxide, organic acids, peptides, or bacteriocins [41]. The above studies indicate that these LAB species have different functions, so it is also worth investigating and exploring the specific actions of strains obtained in this study.

After adhering to the intestinal tract of the organism, LAB can inhibit microorganisms by secreting organic acids, polysaccharides, and antimicrobial peptides. Evivie et al. [42] found that *L. delbrueckii* subsp. *bulgaricus* KLDS 1.0207 could produce antimicrobial peptides that inhibit *Escherichia coli* ATCC25922^T^ and *Staphylococcus aureus* ATCC25923^T^. The antimicrobial properties of LAB metabolites have been widely demonstrated, but the inhibitory effect varies among different strains, and results of the present study demonstrate this point. *L. reuteri* P7, *L. amylovorus* P8, *L. johnsonii* P15 and *L. reuteri* P16 in this study all showed extremely strong inhibition against *Escherichia coli* ATCC 11775^T^ with an inhibition zone diameter of more than 18.00 mm, and moderate ability against *Staphylococcus aureus* ATCC 6538^T^ (>14.00 mm), *Bacillus subtilis* ATCC 19217^T^, *Micrococcus luteus* ATCC 4698^T^, and *Salmonella enterica* ATCC 43971^T^; among these, P15 had an inhibition zone diameter of at least 18.00 mm against all pathogenic strains. The increase in antibiotic-resistant microorganisms and the future inefficiency of current therapies highlight the need to find alternative strategies [43]; on the other hand, the interest in food biopreservation has increased considerably due to the adverse effects of chemical preservatives on health [44]. Therefore, bacteria with antimicrobial activity, such as strains the obtained in this research, can potentially be used in medicine and food.

According to previous studies, LAB must multiply to a certain order of magnitude in order to function in the intestinal tract, and tolerance to the gastric environment with pepsin and pancreatic enzymes in the intestinal fluid is critical to their entry into the intestine to function [45,46]. In the present study, four lactobacilli strains showed different decreases in viability after being exposed to simulated GIT conditions for 7 h. The viable count of P7 and P15 exhibited no significant decrease in SGF and there was a marked drop of P8 and P16, but the survival of P7 significantly decreased in SIF after 4 h. These results are similar to those of Kaewnopparat et al. [47], who reported a significant decrease in the survival rate of *L. fermentum* SK5 in SGF after 3 h to only 70.48%, but exposure to SIF after 4 h had virtually no effect on viable cell numbers. Consequently, the results in this research indicate that the selected strains were tolerant to GIT.

LAB have always been considered safe; however, due to the inappropriate use of antibiotics, an increasing number of LAB are resistant or carry resistance genes, most of which are located on mobile non-chromosomal genetic factors, including plasmids, transposons, integron–gene cassette systems, phages, etc., which can be transferred horizontally to other intestinal strains or even pathogenic bacteria, leading to the creation of superbugs and to increased antibiotic resistance and use. The failure rate of treatment poses serious health risks to animals, which means safety evaluation of LAB is necessary [48]. It has been reported that lactobacilli are usually susceptible to penicillin antibiotics (ampicillin and piperacillin) and *β*-lactamase inhibitors, which are commonly used to inhibit cell wall synthesis, and to antibiotics that inhibit protein synthesis, such as erythromycin and clindamycin [49]. Lavilla-Lerma et al. [50] tested the resistance of *L. plantarum* isolated from Spanish goat cheese and found that all strains were resistant to methomyl, ciprofloxacin, vancomycin, and ticlopidine; about 30% of strains were resistant to penicillin, tetracycline, rifampicin, and levofloxacin; and more than 90% were susceptible to ampicillin, gentamicin, streptomycin, erythromycin, and chloramphenicol. The four selected LAB isolates in the present study displayed susceptibility to carbenicillin, cefamezin, clindamycin, and chloramphenicol, and resistance to amikacin and norfloxacin; additionally, none of the strains showed hemolytic activity, which further confirms the complex mechanism of safety properties in lactobacilli.

The reproductive performance of sows and the growth performance of piglets determine the economic benefits of pig breeding. In research and production, a variety of nutritional control measures are adopted to improve such performance [51]. Pregnant sows are subjected to many stressors during gestation and lactation, and some studies have indicated that feed supplementation with LAB improved reproductive performance in sows. Betancur et al. [52] reported that feed supplementation with *L. plantarum* CAM6 during gestation and lactation decreased piglet mortality and increased litter weight at birth. In this study, LAB supplementation of feed improved sow performance in terms of the number of piglets at birth and weak piglets, birth weight per litter, and conception rate during estrus. The sow’s nutrition not only affects her own health, but also the piglets’ newborn weight, survival rate, weaning weight, and later growth performance, as well as the time to slaughter and the production efficiency of fattening; therefore, adding selected LAB in the present research to positively affect the reproductive performance of perinatal sows is meaningful [53].

How to improve the growth performance and health of piglets has become one of the focuses of concern in the current pig breeding industry, and the growth performance and health of piglets has a direct impact on economic performance. For the host, the effect of homologous strains is better than heterologous, and feeding lactobacilli to piglets at weaning can improve intestinal development, regulate intestinal transport channels and closely linked protein-related gene expression, increase the concentration of volatile fatty acids and lactic acid, reduce intestinal pH, promote the proliferation of beneficial intestinal bacteria such as LAB and clostridium, and reduce the number of pathogenic bacteria as *Escherichia coli*, thus promoting intestinal health and alleviating weaning stress [54,55]. Nordeste et al. [56] showed that adding *L. acidophilus* to the diet could effectively improve the production performance and reduce the diarrhea rate of weaned piglets. Casas et al. [57] showed that *Clostridium butyricum* could improve the growth performance of weaned piglets.

The diets supplemented with combinations of *L. reuteri* P7, *L. amylovorus* P8, and *L. johnsonii* P15 strains in this research showed strong resistance to ETEC K88, high antimicrobial activity, and good adhesion ability, as well as antibiotic effects on the growth performance and diarrhea rate of weaned piglets during the entire experimental period. The LAB group of weaned piglets had significantly increased final body weight (*p* < 0.05) compared to the control and antibiotic groups, and the incidence of diarrhea was lower in the LAB (*p* < 0.05) and antibiotic (*p* = 0.071) groups compared to control, which indicates that selected LAB could improve the growth performance and reduce the diarrhea rate of piglets similar to antibiotics. Compared with the control group, in the LAB and antibiotic groups, ADG, ADFI, and the efficiency of feed utilization were significantly increased (*p* < 0.05). Most important, there was no significant difference between LAB and antibiotic treatment group in the study, which was in agreement with the results of Giang et al. [58], who found that, with the addition of LAB complexes to the diet, ADFI and F:G in weaned piglets were significantly better than in the control group. This study investigated the effect of replacing a full course of antibiotics with LAB complex bacteriological agents on piglet growth, which provides an important reference with regard to the effects of LAB replacement of antibiotics on pig growth performance and the development of future growth programs.

The serum biochemical index is a comprehensive reflection of the body’s metabolic function; that is, changes in this index can reflect alterations of metabolic function. The antioxidant capacity of an organism is reflected by its T-AOC, which is composed of an enzyme-promoted antioxidant system (GSH-Px, SOD, and other enzymes) and a non-enzymatic antioxidant system, and there is a strong correlation between the strength of the T-AOC and the degree of health. The level of MDA, the end product of lipid peroxidation, reflects the degree of oxidative damage in the organism. Several studies have shown that the application of certain LAB strains could improve the antioxidant capacity of pigs [3,59]. Regarding the effect of LAB on T-SOD, T-AOC, and MDA in the serum of sows, this study shows that the MDA content in the serum was significantly lower in the LAB group than the control group. These results are consistent with Boontiam et al. [60], who showed that the MDA concentration of sow serum was significantly lower in the hydrolyzed yeast group than the control group. Although there may be oxidant stress during gestation and lactation, supplementation with LAB improved the antioxidant capacity of the sow. Regarding the effect of LAB on T-SOD, T-AOC, and MDA in the serum of weaned piglets, this study shows that the T-SOD content in the serum was remarkable higher, whereas MDA concentration was significantly lower in the LAB and antibiotic treatment groups than in the control group fed a basal diet. In addition, there was no marked difference between the LAB and antibiotic treatment groups. Compared with the control group, the fermentation broth increased the activity of T-SOD and T-AOC and reduced the MDA content in serum.

Probiotics and their metabolites can act within the host’s immune system and increase the body’s resistance to infection, cancer and allergies. Immunoglobulin and immune cytokines are the main components of humoral immunity, and their quantity can reflect the host’s resistance to pathogenic bacteria, as well as being an important indicator of immune function [61]. It was reported that, compared with the control group, different doses of lactobacilli could increase the content of IgA and IgG in the serum of mice. The content of IgA and IgG in adult mice with protein energy malnutrition decreased, but the contents of both increased after the ingestion of *L. johnsonii* La1, which indicated that La1 was beneficial to host health [62]. *L. casei* supplementation can improve the survival and anti-infection ability of animals, repair the immune response to *Candida albicans*, recruit and activate phagocytes and release proinflammatory cytokines [63]. Several studies have proven that feed supplemented with LAB could improve the serum immunity of pregnant sows and weaned piglets. Laskowska et al. [64] showed that the use of probiotic preparations as feed additives resulted in increased concentrations of the pro-inflammatory cytokines TNF-α and IgA in sows; Mizumachi et al. [65] found that fermented liquid diet feed prepared with the use of *L. plantarum* LQ80 could enhance the immune response by increasing the content of TNF-α and IgG in piglet serum. This study shows that the TNF-α and IgA content in the serum was significantly higher in the LAB group than the control group, and these results are consistent with those of previous methods. In addition, there are no significant differences in IgG content between the LAB and antibiotic group in piglets, as the contents of both are higher than in the control group.

## 5. Conclusions

Under the conditions of this study, 295 selected LAB strains were isolated from fecal samples from 55 healthy weaned piglets, among which *L. reuteri* P7, *L. amylovorus* P8, and *L. johnsonii* P15 showed good inhibition against ETEC K88 and had excellent probiotic properties. Feed supplementation with these three strains could improve the reproductive performance of pregnant sows, as well as the antioxidant capacity and immune indexes of serum, and including LAB in diets for weaned piglets could contribute to improved growth performance, antioxidant capacity and immune indexes in sera, alongside decreasing the incidence of diarrhea during the 4-week post-weaning period. Therefore, *L. reuteri* P7, *L. amylovorus* P8, and *L. johnsonii* P15 might be considered as potential antibiotic alternatives for further study in pregnant sows and weaned piglets.

## Figures and Tables

**Figure 1 animals-11-01719-f001:**
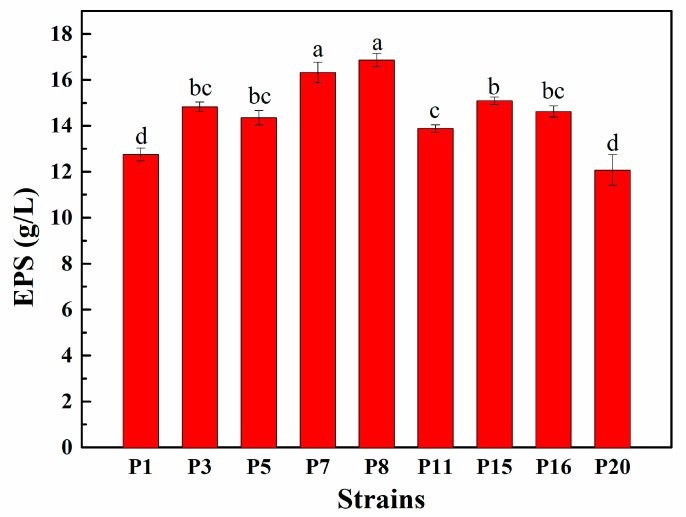
The EPS production capacity of representative LAB isolates. Different lowercase letters (a, b, c, d) denote significant differences (*p* < 0.05), and results were expressed as mean ± standard deviation (SD) (*n* = 3).

**Figure 2 animals-11-01719-f002:**
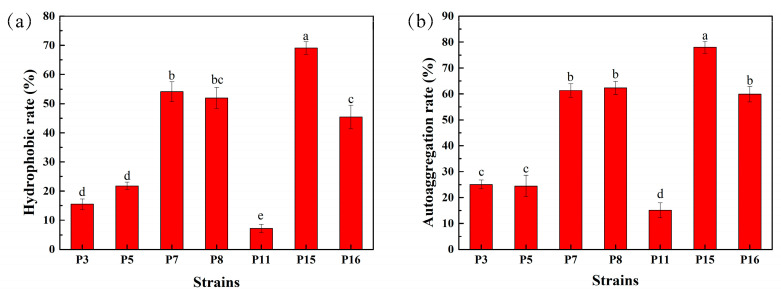
Cell surface hydrophobicity and auto-aggregation ability of representative LAB isolates. (**a**) Hydrophobicity (%), (**b**) auto-aggregation (%). Different lowercase letters (a, b, c, d, e,) denote significant differences (*p* < 0.05), and results were expressed as mean ± standard deviation (SD) (*n* = 3).

**Figure 3 animals-11-01719-f003:**
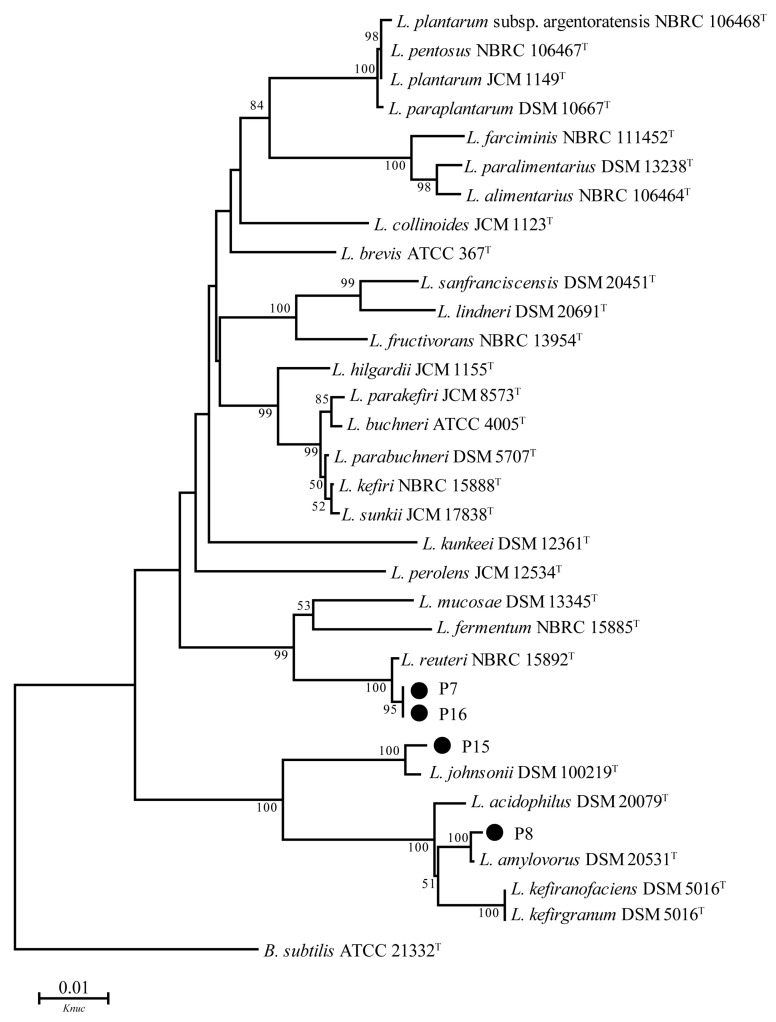
Phylogenetic tree for the selected LAB strains. Phylogenetic tree showing the relative positions of selected isolates P7, P8, P15 and P16 by the neighbor-joining method. *Bacillus subtilis* was used as outgroup, bootstrap values shown at the nodes of the tree are 1000 replicates and the bar indicates 1% sequence divergence. *L*. = *Lactobacillus*, *B*. = *Bacillus*, *Knuc* = nucleotide substitution rate.

**Figure 4 animals-11-01719-f004:**
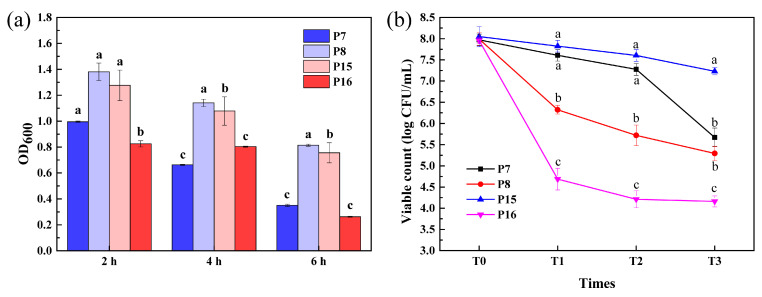
Survival of selected LAB strains in bile salt and simulated gastrointestinal fluids. (**a**) Biomass of four LAB isolates after 0.5% bile salt treatment, (**b**) viable count (log CFU/mL) of selected LAB strains after simulated GIT. Gastric juice T0 = viability at the beginning of gastric juice, T1 = viability after simulation of gastric conditions; intestinal juice T2 = viability at the beginning of gastric juice, T3 = viability after simulation of enteric conditions. Different lowercase letters (a, b, c) on the same row denote significant differences (*p* < 0.05) during the assay, and results were expressed as mean ± SD (*n* = 3).

**Figure 5 animals-11-01719-f005:**
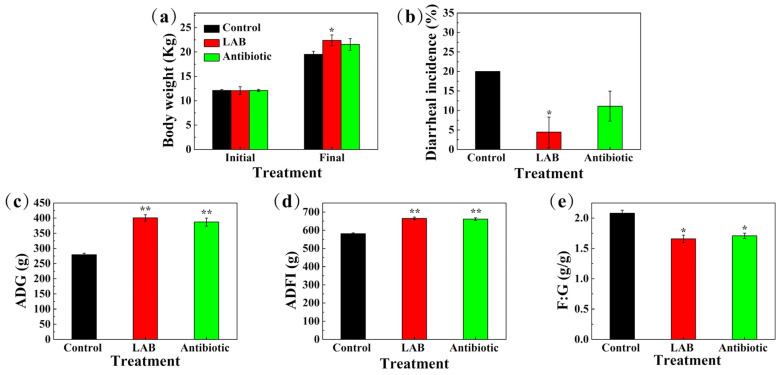
Effects of selected LAB strains and antibiotic on growth performance in weaned piglets. Control group, basal diet (*N* = 30); LAB group, basal diet + 6% mixed selected LAB strains (N = 30); antibiotic group, basal diet + 150 mg/kg of aureomycin (*N* = 30). (**a**) Body weight (Kg), (**b**) diarrheal incidence (%), (**c**) ADG (g), (**d**) ADFI (g) and (**e**) F:G (g/g). ADG, average daily gain; ADFI, average daily feed intake; F:G, feed:gain ratio. Asterisks denote significant differences compared with the control group (* *p* < 0.05, ** *p* < 0.01).

**Figure 6 animals-11-01719-f006:**
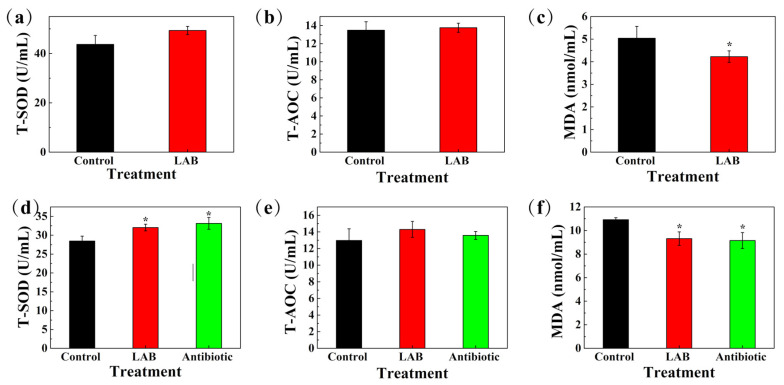
Effects of LAB on antioxidant capacity of serum in sows and weaned piglets. (**a**,**d**) T-SOD (U/mL), (**b**,**e**) T-AOC (U/mL), (**c**,**f**) MDA (nmol/mL), are antioxidant capacity of serum in sows and weaned piglets, respectively. For sows, control, basal diet (*N* = 18); LAB, basal diet + 6% mixed selected LAB strains (*N* = 18). For piglets, control, basal diet (*N* = 30); LAB, basal diet + 6% mixed selected LAB strains (*N* = 30); antibiotic, basal diet + 150 mg/kg of aureomycin (*N* = 30). T-SOD, total superoxide dismutase; T-AOC, total antioxidant capacity; MDA, malondialdehyde. Asterisks denotes significant differences compared with the control group (* *p* < 0.05).

**Table 1 animals-11-01719-t001:** Ingredient and nutrient levels of basal diet in each stage.

Items	Gestation (%)	Lactation (%)	Piglet (%)
Corn	57.34	60.36	58.00
Soybean meal	12.80	23.20	25.00
Fish meal	0	3.00	4.00
Dried whey	4.00	4.50	5.00
Wheat bran	22.10	5.00	3.00
Limestone	1.50	1.40	1.00
CaHPO_4_	0.96	0.99	2.00
NaCl	0.40	0.40	0.30
L-Lysine HCl	0.15	0.30	0.35
DL-Methionine	0	0.15	0.10
Choline chloride	0.15	0.15	0.20
Threonine	0.05	0	0.05
Vitamin-mineral premix ^a^	0.55	0.55	1.00
Total	100.00	100.00	100.00
Nutrient levels			
Digestible energy (MJ/kg)	12.11	13.59	14.00
Crude protein (%)	14.66	17.76	18.50
Ca (%)	1.05	1.08	0.80
Lysine (%)	0.73	1.11	1.35
Methionine (%)	0.40	0.56	0.47

^a^ Premix supplied per kg for sows: 25,000 IU vitamin A, 4500 IU vitamin D3, 20 IU vitamin E, 2.5 mg vitamin K, 8 mg vitamin B2, 1 mg vitamin B1, 8 mg vitamin B11, 4 mg vitamin B6, 0.08 mg biotin, 0.015 mg vitamin B12, 200 mg choline, 12 mg pantothenic acid, 3 mg folic acid, 20 mg nicotinic acid, 150 mg Fe (FeSO_4_), 20 mg Cu (CuSO_4_), 40 mg Mn (MnO), 150 mg Zn (ZnO), 10 mg I (KI), 0.5 mg Se (Na_2_SeO_3_). Premix supplied per kg for piglets: 6000 IU vitamin A, 200 IU vitamin D3, 40 IU vitamin E, 2 mg vitamin K, 200 mg choline, 8 mg pantothenic acid, 3 mg vitamin B2, 3 mg folic acid, 25 mg nicotinic acid, 6 mg vitamin B11, 6 mg vitamin B6, 0.08 mg biotin, 0.01 mg vitamin B12, 100 mg Fe (FeSO_4_), 10 mg Cu (CuSO_4_), 45 mg Mn (MnO), 100 mg Zn (ZnO), 100 mg I (KI), 2 mg Se (Na_2_SeO_3_).

**Table 2 animals-11-01719-t002:** Antimicrobial activities to ETEC K88 of representative LAB isolates.

Isolates	Antimicrobial Activity	Isolate	Antimicrobial Activity
P1	+++	P11	++++
P2	++	P12	++
P3	+++	P13	++
P4	+++	P14	+++
P5	++++	P15	++++
P6	++	P16	++++
P7	+++	P17	+++
P8	++++	P18	+++
P9	++	P19	++
P10	++++	P20	+++

++, diameter of inhibition zone 14.00–18.00 mm; +++, 18.00–22.00 mm; ++++, more than 22.00 mm; the diameter of inhibition zone included that of a hole puncher (10.00 mm).

**Table 3 animals-11-01719-t003:** Physiological and biochemical characteristics of representative LAB isolates.

Isolates	Temperature (°C)					NaCl (*w*/*v*, %)		pH						
	4	10	40	45	50	3.0	6.5	3.0	3.5	4.0	4.5	8.0	9.0	10.0
P1	w	+	+	+	-	+	+	-	w	+	+	+	+	+
P3	w	+	+	+	w	+	+	w	w	+	+	+	+	+
P4	-	w	+	w	-	+	+	-	-	w	+	+	+	w
P5	w	+	+	+	w	+	+	-	w	+	+	+	+	+
P7	w	+	+	+	-	+	+	w	+	+	+	+	+	+
P8	w	+	+	+	+	+	+	-	w	+	+	+	+	+
P10	-	w	+	w	-	+	+	-	w	+	+	+	+	w
P11	w	+	+	+	-	+	+	w	+	+	+	+	+	+
P14	-	w	+	+	+	+	w	-	-	w	+	+	+	+
P15	w	+	+	+	w	+	+	+	+	+	+	+	+	+
P16	w	+	+	+	+	+	+	+	+	+	+	+	+	+
P17	-	+	+	+	-	+	w	-	-	w	+	+	+	+
P18	-	w	+	w	-	+	+	-	w	+	+	+	+	w
P20	w	+	+	+	-	+	+	-	w	+	+	+	+	+

All representative strains were positive for Gram stain and negative for catalase reaction. For fermentation type, all selected strains were homo-fermentative except P10, which was hetero-fermentative. +, could grow; -, could not grow; w, grew weakly.

**Table 4 animals-11-01719-t004:** Antimicrobial spectrum of selected LAB strains.

Isolates	Indicator Bacteria						
	*E.* *coli*	*P.* *aeruginosa*	*S.* *aureus*	*B.* *subtilis*	*L.* *monocytogenes*	*M.* *luteus*	*S.* *enterica*
P7	+++	++	++	++	++	++	++
P8	++++	+	++	+++	+++	++	+++
P15	+++	++++	+++	+++	++++	+++	+++
P16	++++	-	++	++	+	++	++

+, diameter of inhibition zone 10.00–14.00 mm; ++, 14.00–18.00 mm; +++, 18.00–22.00 mm; ++++, more than 22.00 mm; -, no inhibition zone was detected; the diameter of the inhibition zone included that of a hole puncher (10.00 mm). *E. coli*, *Escherichia coli* ATCC 11775^T^. *P. aeruginosa*, *Pseudomonas aeruginosa* ATCC 15692^T^. *S. aureus*, *Staphylococcus aureus* ATCC 6538^T^. *B. subtilis*, *Bacillus subtilis* ATCC 19217^T^. *L. monocytogenes*, *Listeria monocytogenes* ATCC 51719^T^. *M. luteus*, *Micrococcus luteus* ATCC 4698^T^. *S. enterica*, *Salmonella enterica* ATCC 43971^T^.

**Table 5 animals-11-01719-t005:** Antibiotic susceptibility of selected LAB strains.

Isolates	CB	CZ	AM	GM	NOR	CC	P	E	C	AK
P7	S	S	S	I	R	S	I	S	S	R
P8	S	S	R	S	R	S	R	S	S	R
P15	S	S	S	I	R	S	I	S	S	R
P16	S	S	S	S	R	S	R	I	S	R

CB, carbenicillin; CZ, cefamezin; AM, ampicillin; GM, gentamicin; NOR, norfloxacin; CC, clindamycin; P, penicillin; E, erythromycin; C, chloramphenicol; AK, amikacin; S, susceptible; I: intermediate resistant; R, resistant. Created with reference to the latest CLSI (Clinical and Laboratory Standards Institute) standards.

**Table 6 animals-11-01719-t006:** Effects of LAB on the reproductive performance of sows.

Items	Treatment		(Control vs. LAB) *p*-Value
	Control	LAB	
Total born	11.90 ± 0.35	12.30 ± 0.26	0.09
Born alive	11.20 ± 0.10	12.10 ± 0.21	<0.05
Weak piglets	2.95 ± 0.29	2.45 ± 0.37	<0.05
Stillborn piglets	0.27 ± 0.05	0.23 ± 0.02	0.18
Total weight per litter, kg	17.25 ± 0.82	18.45 ± 0.58	<0.05
Individual weight, kg	1.45 ± 0.10	1.50 ± 0.52	0.34
Live litter rate, %	98.32 ± 1.25	98.35 ± 1.31	0.82
Weak litter rate, %	9.82 ± 3.09	7.44 ± 0.78	0.28
Weaning survival rate, %	90.10 ± 1.37	91.50 ± 0.72	0.31
Estrus rate of 1–7 days, %	80.00 ± 8.71	90.00 ± 4.58	0.31
Estrus rate of 8–14 days, %	10.00 ± 1.00	10.00 ± 1.73	1.00
Conception rate during estrus, %	70.00 ± 5.00	100.00 ± 0.00	<0.01

Control, basal diet (Number (*N*) = 18); LAB, basal diet + 6% mixed selected LAB strains (*N* = 18).

**Table 7 animals-11-01719-t007:** Effects of LAB on immune indexes of serum in sows and weaned piglets.

Items	Treatment in Sows		Treatment in Piglets		
	Control	LAB	Control	LAB	Antibiotic
TNF-α, ng/L	121.02 ± 1.34 ^b^	145.41 ± 1.16 ^a^	68.75 ± 0.49 ^b^	107 ± 2.21 ^a^	80.75 ± 1.92 ^b^
IFN-γ, ng/L	59.72 ± 0.87	60.15 ± 0.92	38.17 ± 0.66	39.52 ± 1.03	39.11 ± 0.84
IgG, g/L	9.17 ± 0.69	9.23 ± 0.28	3.97 ± 0.18 ^b^	4.45 ± 0.23 ^a^	4.34 ± 0.10 ^a^
IgM, g/L	2.47 ± 0.32	2.52 ± 0.39	1.15 ± 0.11	1.20 ± 0.05	1.22 ± 0.13
IgA, g/L	1.26 ± 0.09 ^b^	1.76 ± 0.17 ^a^	0.45 ± 0.03 ^b^	0.57 ± 0.13 ^a^	0.48 ± 0.08 ^b^

For sows: Control, basal diet (*N* = 18); LAB, basal diet + 6% mixed selected LAB strains (*N* = 18). For piglets: Control, basal diet (*N* = 30); LAB, basal diet + 6% mixed selected LAB strains (*N* = 30); antibiotic, basal diet + 150 mg/kg of aureomycin (*N* = 30). TNF-α, tumor necrosis factor-α; IFN-γ, interferon-γ; IgG, immunoglobulin G; IgM, immunoglobulin M; IgA, immunoglobulin A. ^a,b^ Different letters indicate significant differences (*p* < 0.05) between different treatment groups.

## Data Availability

The 16S rRNA gene sequence of strains P7, P8, P15 and P16 used to support the findings of this study have been deposited in the GenBank repository with accession number MT967372, MT967483, MT967485 and MT967484, respectively. http://www.ncbi.nlm.nih.gov.

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
