# Peer review of "Screening of Lactic Acid Bacteria with Inhibitory Activity against ETEC K88 as Feed Additive and the Effects on Sows and Piglets"

_animals, 2021, doi:10.3390/ani11061719_

Round 1
Reviewer 1 Report
After several months, I m happy to say that the manuscript now fulfills the requirements for publication. I appreciate the efforts to improve the paper and the changes introduced therein.
Only a minor point remains. The authors should better describe the methodology for immune parameters evaluation and the name of the kits used as reagents in the corresponding section.
Author Response
Comments and Suggestions for Authors
After several months, I’ m happy to say that the manuscript now fulfills the requirements for publication. I appreciate the efforts to improve the paper and the changes introduced therein.
Only a minor point remains. The authors should better describe the methodology for immune parameters evaluation and the name of the kits used as reagents in the corresponding section.
Thanks for the good suggestion. We have added the methodology for immune parameters evaluation and the name of the kits, please see P6, L251- L257.
Serum antioxidant parameters including total superoxide dismutase (T-SOD), total antioxidant capacity (T-AOC) and malondialdehyde activity (MDA) were measured using methods of Hydroxylamine, ABTS and TBA, serum immune indexes as tumor necrosis factor-α (TNF-α) and interferon-γ (IFN-γ) were both analyzed by ELISA method, and immunoglobulins (IgG, IgM and IgA) were determined by Turbidimetric method, respectively. All assays were performed by kits provided by Nanjing Jiancheng Bioengineering Institute (Nanjing, China).

Reviewer 2 Report
REVIEW
Dear authors,
Nowadays, antibiotic resistances are increasing, and alternatives to antibiotics in animal’s diets must be studied. Among this alternatives, probiotics have gained considerable attention with respect to their beneficial effects on livestock performance and health. The most significant effects of probiotics on the gastrointestinal microbiota take place when they are included in diets during some stressful periods such as weaning and/or at the beginning of the lactation period. For that reason, the topic of this study is of a great interest because this work aimed to develop an effective probiotic lactic acid bacteria (LAB) from piglet feces and in vitro characterization of probiotic properties and in vivo performance of sows and weaned piglets in order to be used in animal`s diet. However, in order to be considered for publication some major revisions are necessary.
Overall, the paper needs a thorough and hard edit for English language and usage throughout, especially in the discussion section. The authors should consider having a professional editing service work on the manuscript. Moreover, in the discussion section there are several paragraphs and sentences without any reference. Additionally, a better review of the topic must be made in order to discuss the results obtained in this study because there are several manuscripts that were not included and could support better the results showed.
Line 90. Please indicate how fresh faecal samples from healthy weaned piglets were collected. The material used, if they were collected and transported in any transport medium to the lab…
Line 92. Please indicate the dilution was plated in order to isolate these LAB.
Line 94. Why LAB were isolated at 30ºC instead of 37ºC? Most LAB, especially Lactobacillus spp. from faeces grow at 37ºC. Moreover, all the other in vitro test were made at 37ºC.
Line 96. Please, indicate the morphological shape f these gram positive bacteria isolated.
Line 100. The bibliographic reference of the well diffusion technique must be indicated. Moreover, it must be indicated the concentration of both the LAB and the ETEC K88 used in this in vitro assay.
Line 101-102. It is indicated that LAB strains with relatively larger inhibition zone diameters were selected. However, it must be also indicated compared to what. How many mm is considered to inhibit the pathogen? Did the authors use any inhibition control? What quality control organisms or measures did you use for the antimicrobial susceptibility testing?
Line 105-109. I'd rephrase this paragraph. There is any point separating phrases so it is difficult to understand it. Maybe authors could separate the techniques with a point (line 108, and line 109).
Line 105-110. Authors must indicate for these physiological and biochemical characteristics of selected LAB isolates the concentration of LAB used and how they compared the growth. They must know if LAB grow more or less after the challenge. Or did the authors only measure the growth after the challenge? How the authors know if there is a growth or not?
Line 111. EPS must be explained before its abbreviation in the tittle because it is the first time it is mentioned in the work.
Line 142. What culture medium was used for this technique to allow the growth of all these bacteria? It must be indicated if all well diffusion techniques were made in the same culture medium in order to allow the growth of all these pathogens. Also it must be indicated the concentration of LAB tested in each well and if it was the same concentration for all LAB selected in this step. Also the concentration of the indicator bacteria inoculated to assess the antimicrobial activity of the selected LAB must be indicated.
Line 144. The authors indicate that the inhibition spectrum and inhibition ability of each LAB strain were determined according to the inhibition zone diameter, and strains with relatively larger inhibition zone diameters were selected for further study. However they haven’t use any inhibition halos control. Why? How they can know that these indicator bacteria are sensitive or resistant or weakly sensitive to the antimicrobial activity of the LAB strains selected if there is not compared at the same time with any inhibition halos control at the same time? what quality control organisms or measures did you use for the antimicrobial susceptibility testing?
Line 148. optical density (OD)
Line 151. What modifications? Indicate them.
Line 153. Rephrase to Asessment of antibiotic susceptibility profile of selected LAB.
Line 154. Why were these antimicrobials chosen?
Line 159. Authors need to provide what they used as zone diameters for interpretation (and references).
Line 164-183. Please, indicate also the sanitary status of the farm and the vaccination program in this sub-section.
Line 166. Indicate the use protocol number Id provided by the committee. Also indicate if there is any guidelines of the Country Policy for Animal Protection about the protection of animals used for research.
Line 166. Are all sows from the same line or breed? Indicate it.
Line 171. How these LAB concentration were prepared for diet administration and how were they administrated? Were cultures administrated in form of lyophilized powder or live cells?
Line 172. Authors said in line 172 that the trial lasted 55 days until the end of lactation. However, in line 169 it was said that the basal diet was administrated in separate formulas during pregnancy and lactation. So the LAB were not administrated during the whole gestation period. Please clarify the trial and if the LAB supplementation were at the beginning, in the middle or at the end of the pregnancy and then during the whole lactation period.
Line 172-173. Please indicate the temperature and relative humidity of the farrowing room.
Line 175. Why authors have made a third group for piglets (antibiotic group) and they didn't make this third group for sows?
Line 214. Also 3 piglets from each replicate? Please indicate it.
I recommend to unify M&M titles with Results titles. They must be the same for all headers and they must be presented in the same order in both M&M and results.
Images resolution in results sections are poor. Please attach figures with a better resolution. They are difficult to interpret.
Line 258. Indicate in M&M Indicate in M&M how growth was determined and measured, and the difference according weakly growth and growth (+).
Line 264. Why 13,89g and no any other measure? According to what 13,89g was selected?
Line 288. Add respectively. “L. johnsonii and L. amylovorus species, respectively”.
Line 306. In the title sais antimicrobial and in the table title antibacterial. I recommend to indicate the same.
Line 313. In M&M the eighth point (2.8) was the Survival of selected LAB strains in bile salt and simulated gastrointestinal fluids and in results this is point 3.10 whereas 3.8 is Assessment of antibiotic susceptibility. Please, unify point order.
Line 329. It was abbreviated before (line 148).
Line 330. Why it was not measured at time 0?
Line 370. Why in figure 5, under graph b,c,d and e it is said treatment and no in graph a.
Line 405. Table 7. Why is indicated p value for the treatment in sows whereas treatment in piglets significant differences were expressed with letters?
Line 411. In order to understand better the discussion section it must be re-written following the same order as the results. Moreover, this section needs a thorough and hard edit for English language.
Line 412. Change symptom to clinical sign. A symptom is a manifestation of disease apparent to the patient himself, while a sign is a manifestation of disease that the physician perceives. The sign is objective evidence of disease; a symptom, subjective.
Line 412. Not only ETEC, other enteropathogens, including other bacteria, viruses and parasites can produced infectious diarrhoea.
Line 412-415. This first paragraph is not referenced. Who said that?
Line 415-426. This paragraph is in red colour. The English is not well understood.
Line 417. Add among others after “period.” Because diarrhoea in weaned piglets are also related to other factors, as diet change, stress factors… There are morphological changes in the intestine whose predisposes diarrhoea, also immunological, and enzymatic changes during the weaning period…
Line 418. Change flora to microbiota. Nowadays is more precise to talk about intestinal microbiota than flora.
Line 424. Ig must be indicate what it is before its abbreviation.
Line 425. What is the s from sIgA?
Line 430. Also vaccines are important to control diarrhea.
Line 430-434. This paragraph is not referenced. Who said that?
Line 437. Add method or technique after “well diffusion”.
Line 438. Change study to studies.
Line 447. Is there any article that shows the inhibition activity against E.coli or other enteropathogens of the lactobacilli isolated in your study? P7 and P16 were identified as L. reuteri and P15 and P8 were assigned to L. johnsonii and L. amylovorus species, respectively. Please, review if there is any article who talk about the inhibition activity of this Lactobacillus species isolated in your study.
453-455. this paragraph is not referenced. Who said that?
Line 496. As your strains have been already identified, named them with the genus and the specie name instead of your own classification in the discussion section.
Line 501-506. This paragraph is not referenced. Who said that?
Line 518-524. This paragraph is not referenced. Who said that?
Line 525. Please use susceptible or susceptibility rather than sensitive or sensitivity throughout the manuscript when referring to antimicrobial susceptibility testing and results.
Line 535. The general presence of poor haemolytic activities among LAB is an indication of their safety properties not of drug resistance in lactobacilli.
Line 537-541. This paragraph is not referenced. Who said that?
Line 546-550. This paragraph is not referenced. Who said that?
Line 551-559. This paragraph is not referenced. Who said that?
Line 565. Resistance to what? It must be indicated.
Round 2
Reviewer 2 Report
I appreciate the efforts to improve the paper and the changes introduced therein. I believe the manuscript has been improved and now it fulfills the requirements for publication in Animals.
This manuscript is a resubmission of an earlier submission. The following is a list of the peer review reports and author responses from that submission.
Round 1
Reviewer 1 Report
The paper provides data on a selection process and effects on pig performance of some LAB strains.
English language has to be improved in order to allow a better understanding of the authors' discourse. Moreover, the figures quality is very low so that the reader has difficulties to appreciate the results. I recomment providing TABLES (with LSM and SE or SEM) rather than figures.
L342-346: it is useless to present not significant results. And it would be better to provide all the data in tables.
L396: the assertion would gain to be moderated. Hygiene is probably as important.
L426: Bile salt were not used in this experiment. Why? And thus, what is the interest to discuss this point?
As a rule, the discussion section could be shortened.
Author Response
Response to Reviewer 1 Comments
Point 1: English language has to be improved in order to allow a better understanding of the authors' discourse. Moreover, the figures quality is very low so that the reader has difficulties to appreciate the results. I recomment providing TABLES (with LSM and SE or SEM) rather than figures.
Thank you, we have performed a professional language revision, please see the manuscript, and the proof of revision has been submitted as an attachment.
We have added table 6 (with SEM) to replace figure 5, please see P11, L363; and we also modified the resolution and font size of other figures.
Point 2: L342-346: it is useless to present not significant results. And it would be better to provide all the data in tables.
Thank you for the information. We have deleted the results with no significant difference and provided some data in table as P11, L363, and revised some figures as in P12, L365 and P13, L380 to make the results to be better presented.
Point 3: L396: the assertion would gain to be moderated. Hygiene is probably as important.
Thank you so much, we have revised in P13, L398- 401.
Point 4: L426: Bile salt were not used in this experiment. Why? And thus, what is the interest to discuss this point?
We have determined the biomass of different LAB isolates after treatment in different bile salt environments with different time, and initially, when we submitted the article, we considered that the bile salt experiment was very similar to the simulated gastrointestinal tract experiment, so we didn’t put the results of the bile salt experiment in the article.
We have added the bile salt experiments in P4, L146-149, P10, L325-330 and P11, L3412-346.
Point 5: As a rule, the discussion section could be shortened.
Thank you, we have deleted some contents and references that are less correlated with the present findings in the discussion.

Reviewer 2 Report
The manuscript entitled " Screening of potential probiotic lactic acid bacteria with inhibitory activity against ETEC K88 and their effects on sows and piglets as feed additives" is well organized. The idea and design of this study are good although it is not clear.
However, I suggest the authors should address the following minor comments for further consideration of the manuscript for publication.
The manuscript should be proof read by a native speaker for the correction of the English language so as to meet the standard of publication in the journal.
Check the abbreviations throughout the manuscript and introduce the abbreviation when the full word appears the first time in the text and then use only abbreviation.
The authors did not indicate their ethical agreement number here, please add this piece of information to the material and methods section.
It is general to italicize family, genus, and species, but not name, of viruses on their taxonomy.
Line 208-211. Could the authors provide more information on how body weight and feed consumption were recorded (name of equipment, methodology…)?
- were there losses of piglets during the experiment?
The software package should be included in the statistical analysis section.
The authors should clearly define the protocol used in this experiment and this is applied to the results part
In the discussion, the authors have mentioned many references, but in many part, it is not correlated with the present findings and it is encouraged to includes how the present findings results are observed for the better outcome of the manuscript.
In the discussion, you have generalized the term probiotics instead of using specific action of the tested bacterial species (please mention the name of the bacteria included in the probiotics when you described). Thanks
In Figures 4, 5, and 6. X and Y axis legends and figures were not clear author should correct it.
Author Response
Point 1: The manuscript should be proof read by a native speaker for the correction of the English language so as to meet the standard of publication in the journal.
Thank you, we have performed a professional language revision, please see the manuscript, and the proof of revision has been submitted as an attachment.
Point 2: Check the abbreviations throughout the manuscript and introduce the abbreviation when the full word appears the first time in the text and then use only abbreviation.
Yes, we have checked all abbreviations in the manuscript and revised it.
Point 3: The authors did not indicate their ethical agreement number here, please add this piece of information to the material and methods section.
Thank you. Before the project started, the Life Sciences Ethics Review Committee of Zhengzhou University has reviewed the research content and process of the project, found that it follows the international and national promulgated ethical requirements for biomedical research, thus consent was given for the project to proceed. We have submitted the proof to editor when the paper was initially submitted, and this time we also put the proof in the attachment.
Point 4: It is general to italicize family, genus, and species, but not name, of viruses on their taxonomy.
Thank you for your information, we have italicized family, genus, and species of bacteria on their taxonomy.
Point 5: Line 208-211. Could the authors provide more information on how body weight and feed consumption were recorded (name of equipment, methodology…)?
Yes, we have added more information on body weight and feed consumption recorded, please see P6, L204-208.
Point 6: were there losses of piglets during the experiment?
Yes, there were 7 piglets died during sow production and 9 piglets died during piglet weaning.
Point 7: The software package should be included in the statistical analysis section.
Yes, we have added software package in the statistical analysis section, please see P6, L214-223.
Point 8: The authors should clearly define the protocol used in this experiment and this is applied to the results part
Thank you for your information, we have added clear information on the software and statistical methods, and make sure they are same with the results part, please see P6, L214-223.
Point 9: In the discussion, the authors have mentioned many references, but in many part, it is not correlated with the present findings and it is encouraged to includes how the present findings results are observed for the better outcome of the manuscript.
Thank you, we have deleted some contents and references that are less correlated with the present findings in the discussion.
Point 10: In the discussion, you have generalized the term probiotics instead of using specific action of the tested bacterial species (please mention the name of the bacteria included in the probiotics when you described).
Yes, we have corrected probiotics with the name of the bacteria in the discussion as appropriate.
Point 11: In Figures 4, 5, and 6. X and Y axis legends and figures were not clear author should correct it.
Yes, we have added table 6 to replace figure 5 to make the results to be better presented, please see P11, L363; we also revised the font size and resolution of X and Y axis to make figures clearer, please see figures 5 and 6, P12, L365 and P13, L380.

Reviewer 3 Report
Dear authors,
the manuscript you herein present discusses the selection of 7 LAB strains for the evaluation of their performance as as feed additive in sows and piglets.
Although the manuscript is well written it appears more likely as a copy of your previous work on Molecules. The same approach was used and despite the different sources used no particular attention is due to the real application of this LAB. In my opinion, it should be implemented the experimental workflow on animals.
Author Response
Point 1: Although the manuscript is well written it appears more likely as a copy of your previous work on Molecules. The same approach was used and despite the different sources used no particular attention is due to the real application of this LAB. In my opinion, it should be implemented the experimental workflow on animals.
Thank you for your information.
In 2016, our research team cooperated with a pig farm of Xinxiang, China, and the purpose was to screen potential probiotic lactic acid bacteria (LAB) strains with inhibitory activity against ETEC K88. Therefore, we designed and conducted the experiment, and selected 3 LAB strains L. reuteri P7, L. amylovorus P8 and L. johnsonii P15 that have excellent inhibitory and growth characteristics for using as additives for pregnant sow and piglet, which were all listed in this manuscript. As agreed, we can publish the results of the study, while these 3 strains belong to the pig farm. After that, we wanted to understand how LAB act in the organs and intestinal tract of the animal, in addition to sow breeding and piglet growth performance, and thus correlate with external performance. So, we screened an excellent LAB strains ZA3 from 1100 LAB strains kept in the laboratory in 2018 (this part has been published on the Molecules), and as an additive on the microbial community and fermentation quality of feed has been finished, the effect of ZA3 on animal intestinal microbes and immune function are also underway. These studies will also be published in this and next year.

Round 2
Reviewer 1 Report
Thank you for your replies. The paper is acceptable as such.
Author Response
Thank you very much for your comments and suggestions.
Best regards.
Reviewer 2 Report
Dear authors
Thank you for addressing the comments. The correction was satisfied. So, I recommend the manuscript for publication
Best regards
Author Response
We would like to extend our appreciation to you for carefully reviewing our paper.
Best regards.
Reviewer 3 Report
Dear authors,
I appreciate your efforts but in my opinion, this kind of manuscript should include other data.
Author Response
Dear reviewer,
Thank you very much for your comments and suggestions.
In this manuscript, 295 lactic acid bacteria (LAB) strains were chosen for their inhibitory activity against ETEC K88, and Lactobacillus (L.) reuteri P7, L. amylovorus P8 and L. johnsonii P15 with good inhibition against ETEC K88 and physiological and biochemical characteristics, excellent exopolysaccharide (EPS) production capacity, hydrophobicity, auto-aggregation ability, survival in gastrointestinal (GI) fluids, lack of hemolytic activity and broad spectrum activity against a wide range of microorganisms, were selected for feeding experiment. The previous study published in Molecules evaluated the probiotic properties of the LAB strains isolated from different sources by determining their inhibitory activities to ETEC K88, and strain ZA3 possessed high hydrophobicity and auto-aggregation abilities, had high survival rate in low pH, bile salt environment and GI fluids, sensitive to ampicillin and resistant to norfloxacin and amikacin, without hemolytic activity, did not carry antibiotic resistance genes, and exhibited broad spectrum activity against a wide range of microorganisms [1].
It is not only our 2 studies mentioned above that share similarities in the screening of LAB with good inhibition capacity against ETEC K88 and excellent physiological and biochemical characteristics, we have reviewed a lot of literature on probiotic screening, which basically involves similar experimental elements and methods. Such as Chen et al., [2] who aimed to scrutinize more bacterial strains from Hu sheep milk with potential probiotic activity by screening LAB strains from Hu sheep milk after antimicrobial activity, hydrophobicity of the bacterial surface, growth at different bile salts test and low pH, antibiotic susceptibility assay, resistance to simulated gastric and intestinal conditions tests; Hong et al. [3] selected and characterized potential probiotic for feed additives by antibacterial activity assay, tolerance to low pH and bile acid, microorganism identification and phylogenetic analysis, antibiotic susceptibility and analysis of the GI model system; Olatunde et al. [4] screened potential probiotic LAB in effluents generated during ogi production by selecting of acid and bile salt-tolerant isolates, molecular identification and test of antimicrobial activity; Ozkan et al. [5] isolated LAB strains from Tulum cheeses during various periods of ripening and evaluated their probiotic potential through bile salts and acid tolerance, tolerance to gastric and pancreatic juices, test of cell surface hydrophobicity and auto-aggregation, antibacterial activity and safety characteristics tests; Rastogi et al. [6] investigated the probiotic ability of 2 L. mucosae strains isolated from donkey milk for the first time, specifically focusing on its GI tolerance, cell surface hydrophobicity and aggregation capacity, antagonistic activity against pathogen, antibiotic susceptibility and hemolytic activity; Sirichokchatchawan et al. [7] accessed functional and safety properties of 5 autochthonous LAB strains from pig feces as potential probiotics for a pig feed supplement through a series of experiments including antibacterial activity against pathogens, cell surface properties, antibacterial activity, survival of LAB in different concentrations of bile, low pH and under simulated gastric juice; Somashekaraiah et al. [8] characterized probiotic properties of LAB isolated from naturally fermenting product Neera by in vitro tests including tolerance to acids and bile salts, assessment of antibacterial activity, molecular identification, antibiotic sensitivity, cell surface hydrophobicity and auto-aggregation, survival in simulated gastric and safety evaluation; Wang et al. [9] selected and identified LAB probiotic strains with wide-spectrum and highly efficient antimicrobial activity from infant feces, and evaluated their antibiotic resistance, hemolytic activity, physiological and biochemical characteristics, cell surface hydrophobicity and auto-aggregation ability, tolerance to simulated GI conditions; Zeng et al. [10] revealed probiotic potential of L. plantarum isolated from Chinese homemade pickles through antibacterial activity against pathogens, tolerance to GI tract conditions, cell adhesion activity and safety evaluation. In summary, above studies show that these conventional research methods are essential, common and mature for screening probiotic potential LAB strains with excellent physiological and biochemical properties, therefore, methods related to these researches in this manuscript such as with good inhibition abilities and physiological and biochemical characteristics, excellent EPS production capacity, hydrophobicity and auto-aggregation ability, survival in GI fluids, not have hemolytic activity and broad spectrum activity against a wide range of microorganisms, is not only similar to the previous work on Molecules, but also to the studies of many other researchers involved in screening for probiotic LAB.
As for animal testing, in this manuscript, the final 3 selected LAB strains L. reuteri P7, L. amylovorus P8 and L. johnsonii P15 were used as additives for pregnant sow and piglet, and details of how to add and feed are all listed in this manuscript, please see P5, L193-248.
Moreover, this work is our research team cooperated with a pig farm in 2016, while selected strains should belong to the pig farm and the results of the study can be published 3 years after the end of the trial as agreed. After such a long time, it is impossible for us to perform additional experiments on the original animals, and we are sorry it would be a pity. Fortunately, we screened another excellent LAB strain L. plantarum subsp. plantarum ZA3 from 1100 LAB strains kept in the laboratory in 2018 (this part has been published on the Molecules), although the selected methods are similar, L. plantarum subsp. plantarum ZA3 is a very different species from the previous strains L. reuteri P7, L. amylovorus P8 and L. johnsonii P15, and the aims of are also not same. Firstly, ZA3 used as additive on the microbial community and fermentation quality of feed has been finished, and these studies will also be published in this year; secondly, effect of fermented feed prepared by ZA3 in the immune function, organs and gut microbes of the animal, are in progress, and these studies will also be published in this or next year.
We hope that we have opportunity to continue sharing the research results on the affect mechanism of LAB on fermentation feed and feeding on animals.
Best regards.
References
- Wang, W.; Ma, H.; Yu, H.; Qin, G.; Tan, Z.; Wang, Y.; Pang, H. Screening of Lactobacillus plantarum subsp. plantarum with potential probiotic activities for inhibiting ETEC K88 in weaned piglets. Molecules 2020, 25, 4481. doi:10.3390/molecules25194481.
- Chen, T.; Wang, L.; Li, Q.; Long, Y.; Lin, Y.; Yin, J.; Zeng, Y.; Huang, L.; Yao, T.; Abbasi, M.N., et al. Functional probiotics of lactic acid bacteria from Hu sheep milk. BMC Microbiol. 2020, 20, 228. doi:10.1186/s12866-020-01920-6.
- Hong, S.W.; Kim, J.H.; Bae, H.J.; Ham, J.S.; Yoo, J.G.; Chung, K.S.; Oh, M.H. Selection and characterization of broad-spectrum antibacterial substance-producing Lactobacillus curvatus PA40 as a potential probiotic for feed additives. Anim. Sci. J. 2018, 89, 1459-1467. doi:10.1111/asj.13047.
- Olatunde, O.O.; Obadina, A.O.; Omemu, A.M.; Oyewole, O.B.; Olugbile, A.; Olukomaiya, O.O. Screening and molecular identification of potential probiotic lactic acid bacteria in effluents generated during ogi production. Ann. Microbiol. 2018, 68, 433-443. doi:10.1007/s13213-018-1348-9.
- Ozkan, E.R.; Demirci, T.; Ozturk, H.I.; Akin, N. Screening Lactobacillus strains from artisanal Turkish goatskin casing Tulum cheeses produced by nomads via molecular and in vitro probiotic characteristics. J. Sci. Food Agr. 2020. doi:10.1002/jsfa.10909.
- Rastogi, S.; Mittal, V.; Singh, A. In vitro evaluation of probiotic potential and safety assessment of Lactobacillus mucosae strains isolated from Donkey’s lactation. Probiotics Antimicro. 2019, 12, 1045-1056. doi:10.1007/s12602-019-09610-0.
- Sirichokchatchawan, W.; Pupa, P.; Praechansri, P.; Am-In, N.; Tanasupawat, S.; Sonthayanon, P.; Prapasarakul, N. Autochthonous lactic acid bacteria isolated from pig faeces in Thailand show probiotic properties and antibacterial activity against enteric pathogenic bacteria. Microb. Pathogenesis 2018, 119, 208-215. doi:10.1016/j.micpath.2018.04.031.
- Somashekaraiah, R.; Shruthi, B.; Deepthi, B.V.; Sreenivasa, M.Y. Probiotic properties of lactic acid bacteria isolated from Neera: A naturally fermenting coconut palm nectar. Front. Microbiol. 2019, 10, 1382. doi:10.3389/fmicb.2019.01382.
- Wang, X.; Wang, W.; Lv, H.; Zhang, H.; Liu, Y.; Zhang, M.; Wang, Y.; Tan, Z. Probiotic potential and wide-spectrum antimicrobial activity of lactic acid bacteria isolated from Infant Feces. Probiotics Antimicro. 2020. doi:10.1007/s12602-020-09658-3.
- Zeng, Y.; Li, Y.; Wu, Q.P.; Zhang, J.M.; Xie, X.Q.; Ding, Y.; Cai, S.Z.; Ye, Q.H.; Chen, M.T.; Xue, L., et al. Evaluation of the antibacterial activity and probiotic potential of Lactobacillus plantarum isolated from Chinese homemade pickles. Can. J. Infect. Dis. Med. Microbiol. 2020, 2020, 1-11. doi:10.1155/2020/8818989.
Round 3
Reviewer 3 Report
Dear authors,
I am sorry to have to point out that the work can be improved and that this advice is not accepted. The screening and evaluation of the effects are carried out correctly;however it remains a descriptive work. Why not look at the innate immunity profile? The levels of Ig or those of INF represent useful tools to draw some real deductions. The serum samples are already in the possession of the authors.